# PCNA Ubiquitylation: Instructive or Permissive to DNA Damage Tolerance Pathways?

**DOI:** 10.3390/biom11101543

**Published:** 2021-10-19

**Authors:** Jun Che, Xin Hong, Hai Rao

**Affiliations:** Department of Biochemistry, School of Medicine, Southern University of Science and Technology, Shenzhen 518055, China; hongx@sustech.edu.cn

**Keywords:** DNA damage tolerance, PCNA ubiquitylation, TLS, TS, pathway choice

## Abstract

DNA lesions escaping from repair often block the DNA replicative polymerases required for DNA replication and are handled during the S/G2 phases by the DNA damage tolerance (DDT) mechanisms, which include the error-prone translesion synthesis (TLS) and the error-free template switching (TS) pathways. Where the mono-ubiquitylation of PCNA K164 is critical for TLS, the poly-ubiquitylation of the same residue is obligatory for TS. However, it is not known how cells divide the labor between TLS and TS. Due to the fact that the type of DNA lesion significantly influences the TLS and TS choice, we propose that, instead of altering the ratio between the mono- and poly-Ub forms of PCNA, the competition between TLS and TS would automatically determine the selection between the two pathways. Future studies, especially the single integrated lesion “i-Damage” system, would elucidate detailed mechanisms governing the choices of specific DDT pathways.

A large amount of DNA lesions occur inside each single cell everyday [1]. Although most of these lesions are removed by excision repair pathways, such as nucleotide excision repair (NER) and base excision repair (BER), it is inevitable that some DNA lesions can persist and interfere with the work of DNA polymerases when the cell enters the S phase. Then, the cell uses the so-called DNA damage tolerance (DDT) mechanism to get around these lesions [2]. DDT includes two bypass pathways: translesion synthesis (TLS) or template switching (TS) [3] (Figure 1). DDT is undoubtedly a critical process to survive various DNA damages. A recent study showed that even a single unrepaired DNA lesion can reduce the survival of a budding yeast cell if the DDT is deficient [4].

DDT is a major source for DNA mutations. TLS is often error-prone, whereas TS is largely error-free [3]. Understanding how cells balance between TLS and TS is important, since it determines the mutation rate. During the past decades, researchers have made significant progress in the understanding of the DDT process [2,3,5]. One key module in DDT regulation is ubiquitylation [6], the covalent modification by a small protein called ubiquitin (Ub). In this post-translation modification process, a target protein is recognized by ubiquitin ligase E3, which then works with ubiquitin-conjugating enzyme E2 and ubiquitin-activating enzyme E1 to decorate the target protein with ubiquitin [7]. The number of ubiquitin attached could be one or multiple; in which case, these ubiquitin molecules are often linked together to form a chain [7].

DDT is governed by two sets of E2/E3 ubiquitin enzymes. Rad6 (E2)/Rad18 (E3) add a single ubiquitin (mono-ubiquitylation) to the proliferating cell nuclear antigen (PCNA) on its K164 residue [6]. The mono-Ub is critical for TLS [8]. A TLS polymerase is recruited to the DNA damage site by binding to the ubiquitylated PCNA using its ubiquitin-binding motif or zinc finger (UBM or UBZ) domains [9]. MMS2 and Ubc13 (the E2 hetero-dimer)/Rad5(E3) can further add K63 conjugated poly-Ub chains on top of the mono-Ub, which, in turn, promotes TS in a yet-undefined mechanism [6,8]. How do cells choose between TLS and TS? One could imagine that the selection of TLS or TS is determined by the ubiquitylation state (mono- or poly-Ub) of PCNA. Alternatively, instead of instructing TLS or TS for a lesion, PCNA modifications are only permissive, but the choice between TLS and TS is determined by other factors. Although the evidence is just emerging, we speculate about the potential mechanisms that could impact the TLS/TS choice.

## 1. The Type of DNA Lesion Determines the Choice between TLS and TS

Does the type of DNA damage affect the selection of specific DDT pathways? The cyclobutane pyrimidine dimer (TT-CPD) and thymine-thymine pyrimidine(6-4) pyrimidone photoproduct (TT(6-4)) are the most common types of lesions induced by UV irradiation [10]. In a recent study, each of the lesions was inserted into a yeast chromosome to determine how it is dealt with by the DDT mechanism [4]. Interestingly, TT(6-4) is bypassed largely (~95%) by the error-free mechanism, whereas Rev1-Polζ (Rev3)-mediated TLS accounts for the remaining 5% of the events [4]. In contrast, TT-CPD is bypassed by TS and Polη-mediated TLS almost equally [4].

Methyl methane-sulfonate (MMS)-induced DNA alkylation damage also blocks DNA polymerases. However, the contribution of TLS or TS to damage resistance is different between MMS and UV treatments. The *mms2*∆ cells defective in TS are sensitive to MMS treatment, while *tls*∆ mutants are not sensitive unless being treated at a very high dose of MMS (>0.02% in a spotting assay) [11]. TLS contributes to MMS resistance at low doses when TS is missing [11,12,13]. These studies suggest that TS is a preferrable pathway or TS can compensate for the absence of TLS in MMS-treated cells. In contrast, *tls*∆ is more sensitive than *mms2*∆ mutant cells at low doses of UV treatments (e.g., 5 J/m^2^ in our observation). The mechanism underlying the cellular selection of the TS and TLS pathways upon different lesion types remains enigmatic.

## 2. Hypothesis: Kinetics of TLS or TS Determines the Choice between TLS and TS

PCNA ubiquitylation upon DNA damage is triggered by the uncoupling of DNA helicase and stalled replicative polymerases [14] and the resultant RPA binding to the ssDNA [15]. While all TT(6-4), TT-CPD and MMS lesions can induce these, it is unclear whether they induce a different ratio of PCNA poly-Ub over mono-Ub and, thus, determine the choice between TLS and TS. Alternatively, the ratio of PCNA poly-Ub over mono-Ub is not so different among different lesions, but the competition between TLS and TS determines their usage instead. If TLS is executed faster than TS at a given lesion, then TLS dominates, and vice versa.

TS could be a fork reversal event at a stalled replication fork (Figure 1) [5], which naturally competes with TLS by fork remodeling. However, increasing evidence suggests that TS takes place mostly at post-replicative ssDNA gaps in budding yeast cells [16,17,18,19]. In this case, the kinetic of TS is likely determined by the nucleolytic expansion of small gaps (both from 5′ and 3′) [20,21] and the binding of RPA and Rad51 (the recombination executor) to ssDNA gaps, which should not be altered drastically with different types of DNA lesions. In contrast, the kinetics of TLS should vary among different types of lesions easily.

## 3. Steps Affect the Kinetics of TLS Potentially

TLS contains two steps: nucleotide(s) incorporation opposite to the lesion site and extension from the incorporated nucleotide(s) and involves two rounds of polymerase switch [22]. Thus, the kinetics of a successful TLS could vary significantly in response to different types of DNA lesions. We speculate that the following steps can affect how fast a TLS could occur (Figure 2). (1) The dissociation of polδ or polε from a lesion by falling off or by proteosome-mediated degradation [23]. Some lesions might look “sticky” to a certain replicative polymerase. (2) The availability and kinetics of TLS polymerases that execute the insertion or extension steps. A TLS polymerase is capable of inserting dNTP opposite to different DNA lesions with different efficiencies and accuracies [22] and can be highly efficient and/or less mutagenic only for a certain DNA lesion (the concept of a cognate lesion) [24]. It is also notable that different TLS polymerases are not temporally or spatially equally available in vivo. The protein levels of Polη and Rev1 fluctuate during the cell cycle [25,26]. Polη is available at a stalled replication fork [27]. It performs TLS efficiently at TT-CPD [28,29] and accounts for a large portion of TLS at a single TT-CPD [4]. However, Polη performs little TLS on a single TT(6-4) [4]. Instead, Rev1-Polζ mediates its TLS [4] and likely occurs very slowly behind the replication fork [30], which could explain why a single TT(6-4) is bypassed mostly by the error-free TS mechanism and occasionally by Rev1-Polζ. In line with these results, when the error-free TS pathway is defective, Rev1-Polζ-mediated TLS is dramatically increased at TT(6-4) [4]. Interestingly, a lack of Polη leads to more TS at TT-CPD instead of being compensated for by Rev1-Polζ [4], likely also due to its slowness. These results are also in agreement with the temporal separation between TS and Rev1-Polζ-mediated TLS, as suggested by the peak levels of Rad5 and Rev1 are in the S and G2 phase, respectively [25,31]. (3) How fast a TLS polymerase can be switched back to a replicative polymerase [22]. (4) Replicative polymerases can inhibit TLS by proofreading [32], and the inhibition is likely relieved by increased dNTP pools in cells [33]. We speculate that if any of these steps are slow, TS will dominate over TLS.

## 4. PCNA Mono-Ub Likely Does Not Instruct Which TLS Polymerase to Be Used

PCNA mono-Ub is critical for recruiting UBM/UBZ domain-containing TLS polymerases to the DNA damage site, but it does not necessarily determine which TLS polymerase to use for a certain lesion. Swaps of UBM/UBZ domains between different TLS polymerases might help testing this assumption. Interestingly, the attachment site of Ub on PCNA can be flexible without reducing the efficiency of TLS significantly [34,35,36], suggesting that TLS polymerases are recruited not through a bipartite interaction using both PIP and UBM/Z domains [37]. This argues against the possibility that PCNA mono-Ub has a different affinity to a variety of TLS polymerases due to the distinct location of the UBM/UBZ domain within the TLS polymerases. The potential mechanism for lesion discrimination and the selection of an appropriate TLS polymerase for a specific lesion remains elusive [24]. If a TLS polymerase is switched onto a lesion but less efficient or inappropriate in the insertion step, this might slow down the TLS process and lead to increased TS on that lesion.

## 5. Stoichiometry and Dynamics of PCNA Ubiquitylation

One PCNA ring consists of three PCNA monomers, and consequently, there are three K164 residues with the potential to be ubiquitylated. If the reactions of TLS and TS can take place simultaneously and only the faster prevail, then mono-Ub and poly-Ub may need to coexist in the same PCNA ring. Alternatively, if only one PCNA monomer is allowed to be modified within the ring at a time, then the ubiquitylation status of the PCNA monomer should be dynamically changed to facilitate TLS or TS. In principle, PCNA can get further polyubiquitylated if TLS is slow, and PCNA might get deubiquitylated to allow TLS. Future works are needed to solve these issues. Indeed, Takahashi and coauthors suggest a possibility that a dynamic assembly and disassembly of PCNA ubiquitylation have an impact on lesion bypass [37]. No matter which one is the case, the PCNA ubiquitylation status can be secondary and is dynamically changed to fit for either TLS or TS, depending on which one is faster or slower.

## 6. Damage Load Impacts on TLS/TS Choice

Interestingly, two proximal DNA lesions located on both the top and the bottom DNA strands, which abrogate TS due to lack of template strand formations, leads to increased TLS in *Escherichia coli* cells [38]. A similar paradigm should still hold true in eukaryotic cells. Indeed, we observed that TLS promotes efficient TS when yeast cells are treated with a higher, but not lower, dose of MMS (unpublished results). This observation suggests the possibility that TLS on a ssDNA gap is necessary for a proximal TS on the opposite strand. It might also explain why the *tls*∆ mutant is sensitive to MMS only at a high dose. Furthermore, if TS is the preferable pathway for lesion bypass during MMS treatment, then poly-Ub might get deubiquitylated to promote TLS when TS is not possible.

## 7. Other Factors Affecting TLS/TS Choice

Other factors, such as the chromatin state (heterochromatin vs. euchromatin), nucleosome position upstream of the blocked 3′ end, might also affect the choice between the TLS and TS. It is worth mentioning that the choice between TLS and TS could not be easily detected in a population of cells treated with bulky DNA damage. Thanks to the single lesion chromosome integration system [4], these questions could be answered in the near future. It would also be interesting to examine how the TLS or TS machinery is regulated in response to environmental or physiological cues.

In summary, we propose a hypothesis that, when a lesion is confronted by a DNA polymerase, the reactions of TLS and TS take place simultaneously; only a more efficient and faster reaction will prevail for a certain lesion. PCNA ubiquitylation (mono-Ub or poly-Ub) is likely only permissive to TLS or TS, respectively, but not the deciding factor for the selection of TLS or TS.

## Figures and Tables

**Figure 1 biomolecules-11-01543-f001:**
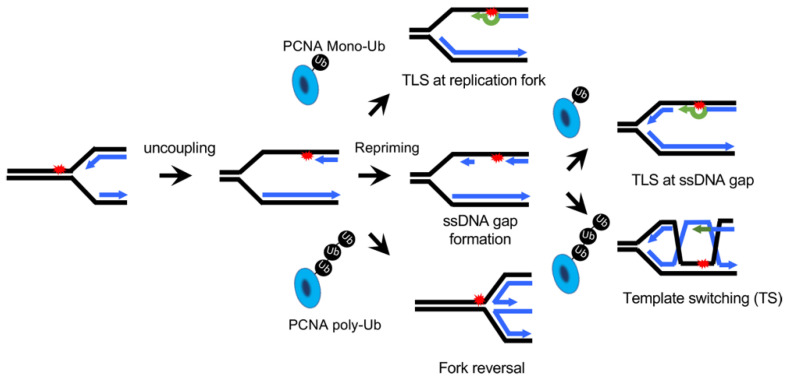
Multiple pathways of DNA damage tolerance (lesion bypass) on the leading strand. A DNA template lesion on the leading strand leads to the uncoupling of the DNA helicase and the replicative polymerase, as well as fork stalling. The lesion can be bypassed right at the stalled replication fork by TLS or, presumably, by fork reversal. Alternatively, the lesion can be skipped by the repriming mechanism, leaving a ssDNA gap behind the replication fork. The ssDNA then can later be filled up by the TLS or TS mechanism.

**Figure 2 biomolecules-11-01543-f002:**
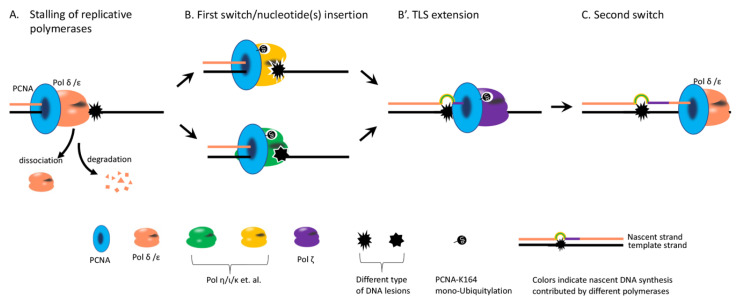
Illustration of the key steps during TLS. (**A**) DNA lesion causes stalling of a replicative DNA polymerase. It leads to dissociation or degradation of the polymerase. (**B**) A TLS polymerase capable of incorporating nucleotide(s) opposite to the DNA lesion is switched on at the blocked 3′ end. PCNA mono-Ub facilitates the recruitment of the “insertion” TLS polymerase. (**B’**) In some cases, a second TLS polymerase (Polζ is very efficient at the extension) extends from the incorporated nucleotide(s) if the first TLS polymerase only performs the incorporation. (**C**) The TLS polymerase is switched back to a replicative polymerase.

## Data Availability

The data presented in this study are available within the article.

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
