# Peer review of "PCNA Ubiquitylation: Instructive or Permissive to DNA Damage Tolerance Pathways?"

_biomolecules, 2021, doi:10.3390/biom11101543_

Round 1

Reviewer 1 Report

This manuscript gives a perspective review on the factors which determines pathway choice between translesion synthesis (TLS) and template switching (TS) to deal with damaged DNA during DNA replication. The authors first discuss the possibility of DNA lesion type in the pathway choice, and hypnotized and analyzed that the kinetics of TLS and TS may determines the choice between TLS and TS, and further introduced the major steps that are closely associated with kinetics of TLS.  In addition, they excluded the involvement of PCNA mono-Ub in selection of TLS polymerase during DNA polymerase switch, with very brief comments.

Overall, the hypothesis presented in the perspective is interesting.

         Major concern:

  1. The authors should discuss a little more about PCNA mono-Ub’s role in DNA polymerase switch. At least they should give more evidence to support their idea that PCNA mono-ub does not instruct DNA polymerase selection.

Minor concerns:

  1. The writing should be improved. And native English speaker may be necessary. The following places should be corrected.
  2. Line 31, in “a “post-translation…, “a” should be “this”.
  3. Line 35 and 37, annotation of E2 and E3 should be consistent.
  4. Line 40, “do” should be added between “how” and “cells”.
  5. Line 40 to 41, the sentence should be rewritten.
  6. Line 51, 52, 74…, “-” should be added between “rev3” and “mediated”. PolyUb or Poly-Ub should be consistent.
  7. Line 65, multiple
  8. Line 81 to 83, the sentence should be rewritten.
  9. Line 85, “be varied” should “vary” as that in Line 96.
  10. Line 86, affecting
  11. Line 97, affect
  12. Line 105, performs
  13. Line 106, mediates
  14. Line 122, affect
  15. Line 129, “biochemical reactions” is not a good description here.
  16. Line 131 to 132, the last sentence should be rewritten.

Author Response

This reviewer found our hypothesis interesting and raised one major concern.

  1. The authors should discuss a little more about PCNA mono-Ub’s role in DNA polymerase switch. At least they should give more evidence to support their idea that PCNA mono-ub does not instruct DNA polymerase selection.

Reply: The reviewer’s suggestion is reasonable, and we have added more discussion/speculation on this subject. Nevertheless, the actual evidence is lacking. To the best of our knowledge, this area of research remains highly speculative. Our hypothesis raises an interesting possibility.

This reviewer has raised a few minor concerns:

  1. The writing should be improved. And native English speaker may be necessary.

Reply: We understand the reviewer’s concern and have edited the manuscript throughout with the help of a native speaker. 

 The following places should be corrected.

Line 31, in “a “post-translation…, “a” should be “this”.

Reply: We agree and have made the change.

  1. Line 35 and 37, annotation of E2 and E3 should be consistent.
  2. Line 40, “do” should be added between “how” and “cells”.
  3. Line 51, 52, 74…, “-” should be added between “rev3” and “mediated”. PolyUb or Poly-Ub should be consistent.
  4. Line 65, multiple 
  5. Line 85, “be varied” should “vary” as that in Line 96.
  6. Line 86, affecting
  7. Line 97, affect
  8. Line 105, performs
  9. Line 106, mediates
  10. Line 122, affect
  11. Line 129, “biochemical reactions” is not a good description here.

Reply:  We appreciate the reviewer’s suggestion and have addressed these issues accordingly.

  1. Line 40 to 41, the sentence should be rewritten.
  2. Line 81 to 83, the sentence should be rewritten.
  3. Line 131 to 132, the last sentence should be rewritten.

             Reply:  We have rewritten these sentences to clarify our views.

Reviewer 2 Report

The perspective manuscript entitled “PCNA ubiquitylation: instructive or permissive to DNA damage tolerance pathways?” speculates about the possibility that DNA Damage Tolerance (DDT) pathway choice is determined by the kinetics of the specific pathways (TLS and TS) rather than the ubiquitylation state of PCNA. The hypothesis is interesting and partially based on the recent manuscript by Maslowska and colleagues, where they analyzed the partitioning of the DDT pathways on a single lesion. To further strength the manuscript, a few points should be added or discussed in the text:

  • Not only the type of lesion seems to affect the selection of the specific DDT pathway but also the damage load. Indeed, the proximity of replication-blocking lesions on opposite strands leads to an increase of TLS in Ecoli since TS cannot operate (Chrabaszcz et al, 2018, NAR). Accordingly, higher dose of the alkylating agent MMS leads to a higher dependency on TLS for ssDNA gap resolution in S. cerevisiae (Wong et al, 2020, Mol Cell).
  • The authors have not discussed about the fact that one PCNA ring consists of three PCNA monomers and, consequently, there are three K164 residues with the potential to be ubiquitylated. This is important since, up to my knowledge, it is unknown whether mono-Ub and poly-Ub can coexist in the same PCNA ring. The authors hypothesize that TLS and TS reactions can take place simultaneously and only the fastest will prevail. Do the authors then imply than mono-Ub and poly-ub coexist on PCNA? Or might PCNA get further polyubiquitylated if TLS is slow? Some authors have also proposed that polyubiquitylation of PCNA inhibits TLS (Takahashi et al, 2020, NAR, and references therein) but, in principle, PCNA might get de-ubiquitylated to allow TLS. In summary, it will be important to discuss about the dynamic of PCNA ubiquitylation.
  • A temporal separation between TS and TLS is suggested by the expression levels of Rad5, which peak in S phase (Ortíz-Bazán et al., 2014, Cell Reports), and Rev1, most abundant in G2/M phase (Waters and Walker, 2006, PNAS).

Minor Points:

  • There are a few typos in the manuscript (for instance, in the legend of Figure 1: “poly-merase” or “mecha-nism”). Please check English language and style.
  • When the authors refer to the article of Maslowska et al., they always mention that just Pol zeta mediates TLS at TT(6-4). However, in this article, it is shown that both Rev3 and Rev1 perform TLS at these lesions.

Author Response

This reviewer found our hypothesis interesting and suggested to add a few points to further strength the manuscript. We really appreciate the reviewer’s insights. Specific issues raised are addressed below.

  1. Not only the type of lesion seems to affect the selection of the specific DDT pathway but also the damage load. Indeed, the proximity of replication-blocking lesions on opposite strands leads to an increase of TLS in Ecoli since TS cannot operate (Chrabaszcz et al, 2018, NAR). Accordingly, higher dose of the alkylating agent MMS leads to a higher dependency on TLS for ssDNA gap resolution in S. cerevisiae (Wong et al, 2020, Mol Cell).

Reply: The reviewer’s point is insightful and well-taken. We have added a new section to discuss about the impact of DNA damage load on DDT choice. In addition, we have brought up some related unpublished data in this section to argue the importance of DNA damage load on the DDT pathway selection.

  1. The authors have not discussed about the fact that one PCNA ring consists of three PCNA monomers and, consequently, there are three K164 residues with the potential to be ubiquitylated. This is important since, up to my knowledge, it is unknown whether mono-Ub and poly-Ub can coexist in the same PCNA ring. The authors hypothesize that TLS and TS reactions can take place simultaneously and only the fastest will prevail. Do the authors then imply than mono-Ub and poly-ub coexist on PCNA? Or might PCNA get further polyubiquitylated if TLS is slow? Some authors have also proposed that polyubiquitylation of PCNA inhibits TLS (Takahashi et al, 2020, NAR, and references therein) but, in principle, PCNA might get de-ubiquitylated to allow TLS. In summary, it will be important to discuss about the dynamic of PCNA ubiquitylation.

Reply: We really appreciate the insightful issues raised. These are great questions and interesting puzzles to be answered in the future. Takahashi and the coworkers’ 2020 NAR paper is insightful. However, it is not clear in his/her paper whether a wild-type copy of POL30 was present while the integration of other pol30 mutants. If both WT and ub*-pol30* are present in the cell, and if they have equal chance to be assembled in a PCNA ring presumably, then the probability of a PCNA ring containing at least one ub* is 7/8 (1-1/2*1/2*1/2). That is relative a large fraction (87.5%) of all PCNA rings in cells.  And if this strain is not sensitive to UV or MMS, then it’d still contain intact TS, which would probably suggest mono- and poly-Ub can co-exist in the same PCNA ring likely. More solid evidences are required to address this issue.

The suggestion of alternative possibility about dynamic regulation of PCNA ubiquitylation is also well-taken. We agree with the comments and have added a new section to discuss about this possibility. 

  1. A temporal separation between TS and TLS is suggested by the expression levels of Rad5, which peak in S phase (Ortíz-Bazán et al., 2014, Cell Reports), and Rev1, most abundant in G2/M phase (Waters and Walker, 2006, PNAS).

Reply:  We agree this indeed could be a contributing factor for why TT(6-4) and TT(CPD) are bypassed differently. We have integrated this point to the section 3 of our revised manuscript. We very much appreciate the insightful comments that improve the manuscript.

Minor Points:

  1. There are a few typos in the manuscript (for instance, in the legend of Figure 1: “poly-merase” or “mecha-nism”). Please check English language and style.

Reply: We appreciate the suggestion and have edited the manuscript extensively.

  1. When the authors refer to the article of Maslowska et al., they always mention that just Pol zeta mediates TLS at TT(6-4). However, in this article, it is shown that both Rev3 and Rev1 perform TLS at these lesions.

Reply: The point is well taken. We have replaced Rev3 with Rev1-Polz .

Reviewer 3 Report

In the manuscript "PCNA ubiquitylation: instructive or premissive to DNA damage tolerance pathways?", the authors describe how mono- or poly ubiquitylation drives the choice between trans-lesion synthesis or error-free template switching.

This article is interesting and important.

It is well written and easy to understand.

This manuscript would be interesting to the scientific community in general.

This reviewer has no concerns with the manuscript as is.

Author Response

This reviewer found our manuscript “interesting and important, well written and easy to understand, would be interesting to the scientific community in general” and had no concerns of our manuscript.

              Reply: we appreciate the reviewer’s positive comments. There is no specific issue to be addressed with this reviewer.